# Rapamycin Affects the Hippocampal SNARE Complex to Alleviate Cognitive Dysfunction Induced by Surgery in Aged Rats

**DOI:** 10.3390/brainsci13040598

**Published:** 2023-03-31

**Authors:** Ning Kang, Xiaoguang Han, Zhengqian Li, Taotao Liu, Xinning Mi, Yue Li, Xiangyang Guo, Dengyang Han, Ning Yang

**Affiliations:** 1Department of Anesthesiology, Peking University Third Hospital, Beijing 100191, China; 2Department of Spine Surgery, Beijing Jishuitan Hospital, Beijing 100035, China; 3Department of Spine Surgery, Peking University Fourth School of Clinical Medicine, Beijing 100035, China; 4Beijing Key Laboratory of Robotic Orthopaedics, Beijing 100035, China

**Keywords:** delayed neurocognitive recovery (dNCR), hippocampal synaptosome, SNARE complex, rapamycin, aged rats

## Abstract

Delayed neurocognitive recovery (dNCR) is a common complication that occurs post-surgery, especially in elderly individuals. The soluble N-ethylmaleimide-sensitive factor attachment protein receptor (SNARE) complex plays an essential role in various membrane fusion events, such as synaptic vesicle exocytosis and autophagosome–lysosome fusion. Although SNARE complex dysfunction has been observed in several neurodegenerative disorders, the causal link between SNARE-mediated membrane fusion and dNCR remains unclear. We previously demonstrated that surgical stimuli caused cognitive impairment in aged rats by inducing α-synuclein accumulation, inhibiting autophagy, and disrupting neurotransmitter release in hippocampal synaptosomes. Here, we evaluated the effects of propofol anesthesia plus surgery on learning and memory and investigated levels of SNARE proteins and chaperones in hippocampal synaptosomes. Aged rats that received propofol anesthesia and surgery exhibited learning and memory impairments in a Morris water maze test and decreased levels of synaptosome-associated protein 25, synaptobrevin/vesicle-associated membrane protein 2, and syntaxin 1. Levels of SNARE chaperones, including mammalian uncoordinated-18, complexins 1 and 2, cysteine string protein-α, and N-ethylmaleimide-sensitive factor, were all significantly decreased following anesthesia with surgical stress. However, the synaptic vesicle marker synaptophysin was unaffected. The autophagy-enhancer rapamycin attenuated structural and functional disturbances of the SNARE complex and ameliorated disrupted neurotransmitter release. Our results indicate that perturbations of SNARE proteins in hippocampal synaptosomes may underlie the occurrence of dNCR. Moreover, the protective effect of rapamycin may partially occur through recovery of SNARE structural and functional abnormalities. Our findings provide insight into the molecular mechanisms underlying dNCR.

## 1. Introduction

Delayed neurocognitive recovery (dNCR) is a frequent neurological complication after anesthesia, surgery, and hospitalization, particularly in the elderly [1], and is characterized by impaired memory, attention, and other cognitive functions. dNCR specifically describes cognitive deterioration occurring from 7 to 30 days post-surgery [1]. Frequencies of dNCR are 23–81% after cardiac surgery and 10–26% after non-cardiac surgery [2,3]. In recent years, dNCR has attracted attention because it causes economic and medical burdens, reduces quality of life, increases morbidity, and causes early mortality [4,5]. dNCR can have a long-term impact on patients’ daily lives by impairing memory, attention, activity, and perception [6]. Unfortunately, because the pathological mechanisms underlying dNCR have not been completely elucidated, there are presently no effective preventive or therapeutic strategies. 

Disruptions in neurotransmitter release play decisive roles in multiple neuropathological events, including dNCR [7]. Key steps of neurotransmitter release include vesicle docking, priming, Ca^2+^-mediated fusion with the presynaptic membrane, and, ultimately, neurotransmitter release from the axon [8]. An important component for this process in presynaptic terminals is the soluble N-ethylmaleimide-sensitive factor attachment protein receptor (SNARE) complex, a four-helix bundle composed of syntaxin 1 and synaptosome-associated protein 25 (SNAP25) on the plasma membrane and synaptobrevin/vesicle-associated membrane protein 2 (VAMP2) on the vesicle membrane. Abnormal SNARE complex structure or function leads to impaired vesicle docking, ultimately resulting in dysfunctional neurotransmitter release [9,10]. Accumulating evidence shows that structural and functional abnormalities of the SNARE complex are related to the occurrence of cognitive dysfunction in the elderly population [11]. 

The SNARE complex is also necessary for mediating membrane fusion in autophagy [12]. Macroautophagy (hereafter referred to as autophagy) is a natural, lysosome-induced process in which intracellular damaged organelles and unnecessary long-lived proteins are degraded; the resulting degradation products are important for maintaining and regulating eukaryotic cell homeostasis [13,14]. The basic morphological progression of autophagy involves phagophore nucleation and elongation, autophagosome formation, and autophagosome–lysosome fusion (autolysosome formation) to degrade unwanted cytosolic materials for recycling. During these processes, tight coordination of vesicle fusions is necessary for both autophagosome biogenesis and their subsequent degradation in lysosomes. Accordingly, SNARE proteins are recognized as key components driving membrane fusion involved in the aforementioned steps of autophagy [12,15,16]. Beclin 1, a core component of the class III phosphatidylinositol 3-kinase complex, is involved in the early stage of autophagosomal membrane formation and is essential for the recruitment of other autophagy-related proteins involved in the expansion step. In addition, the autophagic protein p62/SQSTM1 is selectively incorporated into autophagosomes through direct binding to LC3 and is efficiently degraded by autophagy. Therefore, p62 levels inversely correlate with autophagic activity [17].

In vitro, SNARE complex assembly is inefficient and its uncatalyzed disassembly is undetectable [17]. Therefore, many chaperone molecules are necessary to orchestrate SNARE complex assembly and disassembly (thereby regulating the entirety of neurotransmitter release). These chaperones include mammalian uncoordinated-18 (Munc-18), complexins 1 and 2, cysteine string protein α (CSPα), and α-synuclein.

We previously showed that exploratory laparotomy can cause neurobehavioral deficits that are partially attributable to α-synuclein oligomer aggregation, inhibited autophagy, and disrupted neurotransmitter release in the hippocampal synaptosomes of 22-month-old male Sprague–Dawley rats [7,18]. However, the mechanism by which the SNARE complex is involved has yet to be reported. The present study examined the effects of surgery on SNARE proteins, vesicle-associated proteins, and molecules chaperoning SNARE assembly and disassembly, such as Munc-18, complexins 1 and 2, and CSPα. We also explored whether the SNARE complex participates in the protective effects of rapamycin against cognitive impairment induced by surgery. Our findings broaden the understanding of the molecular mechanisms driving dNCR.

## 2. Materials and Methods

### 2.1. Animals and Ethics

Male Sprague–Dawley rats (22 months old, 550–650 g) provided by the Dongchuang Laboratory Animal Center (Changsha, Hunan, China) were housed in a room with constant temperature (24 ± 1 °C) and humidity (55 ± 5%) under a 12 h light/dark cycle (lights on at 07:00). Rats were allowed to eat and drink freely and were given 1 week to adapt to the new environment before formal experiments. The experimental protocol was approved by the Peking University Biomedical Ethics Committee Experimental Animal Ethics Branch. Utmost efforts were made to minimize the number and suffering of experimental animals.

### 2.2. Anesthesia plus Surgery Protocol 

The anesthesia plus surgery protocol was performed according to our previous report with some modifications [7,18]. Anesthesia was induced by intraperitoneal injection of 50 mg/kg propofol (Diprivan; Astra-Zeneca, London, UK) on a thermal blanket. After loss of the righting reflex, the lateral tail vein was cannulated with a 24G intravenous catheter. Animals were then intravenously infused by springe pump (Mindray, Benefusion VP5, China) with propofol at a dose of 0.9 mg/kg/min. Exploratory laparotomy was performed aseptically on a thermal blanket with a constant temperature of 35 °C. An appropriate area of the abdomen was shaved and sterilized, then a 4 cm vertical incision was made 5 mm below the lower right rib. Approximately 10 cm of the small intestine was exteriorized and vigorously rubbed between the thumb and index finger for 30 s. Next, the small intestine was placed back inside the abdominal cavity and the incision was sutured with surgical staples. The operation lasted 20–25 min and propofol infusion was stopped immediately after the operation. Three milligrams (1 mL) of oxybuprocaine hydrochloride gel was aseptically applied to the surgical site before suturing and subsequently every 8 h for the next 48 h. Treatment of the sham group was identical to that of the surgery group, except laparotomy was not carried out.

### 2.3. Experimental Grouping and Treatment Scheme

Sixty 22-month-old rats were randomly assigned to four groups (n = 15 per group): sham, surgery, sham plus rapamycin (sham+rapamycin), and surgery plus rapamycin (surgery+rapamycin). Rapamycin was dissolved in 25 mg/mL dimethyl sulfoxide and further diluted in a solution containing polyethylene glycol 400 and 5% Tween 80, in accordance with our previous reports [7,18]. Rats in the sham group were induced as described above and received sham surgery. Rats in the surgery group were induced as described above and then received exploratory laparotomy. Rats in the sham + rapamycin and surgery + rapamycin groups were intraperitoneally injected with rapamycin at 10 mg/kg/d for seven consecutive days until the day before the sham operation or exploratory laparotomy. 

### 2.4. Arterial Blood Gas and Blood Glucose Analysis

Blood gas analysis was performed to detect whether hypoxia, hypercapnia, or hypoglycemia occurred during anesthesia and surgery. Immediately after anesthesia was concluded, 1.0 mL blood samples were taken from each rat (n = 5 per group) via cardiac puncture for arterial blood gas analysis (OPTI Medical Systems, Roswell, GA, USA) and blood glucose measurement (Life Scan, Milpitas, CA, USA) [7]. These rats were euthanized after cardiac puncture and not used for any further analyses.

### 2.5. Morris Water Maze Test

Spatial learning and memory status were assessed by a Morris water maze test (Sunny Instruments, Beijing, China) 24 h after sham operation or surgery, in accordance with our previous report [7]. For the place navigation test, rats received four training trials daily for 5 consecutive days to test learning function. On the sixth day, the platform was removed, and each rat was allowed to swim in the pool for 2 min as the probe test to assess memory function. Carbon dust, an edible pigment, was used to darken the swimming water. Swimming was tracked by video (Sunny Instruments) to record and analyze the escape latency, swimming speed, percentage of time spent in the target quadrant, and number of platform crossings of each rat.

### 2.6. Euthanasia and Sample Collection

Immediately after the Morris water maze test was finished, rats were anesthetized in a box prefilled with 6% isoflurane for 60 s and decapitated. Hippocampi were immediately dissected and stored at −80 °C until further processing [19].

### 2.7. Synaptosome Extraction

Synaptosomes were isolated according to our previous report [7]. Hippocampi were homogenized and centrifuged at 10,000× *g* at 4 °C for 10 min. The resulting supernatant was centrifuged at 17,000× *g* at 4 °C for 20 min to obtain the P2 fraction for further Percoll (Pharmacia Biotech, Uppsala, Sweden) gradient separation. Fractions between 10% and 23% were collected for Western blot analysis. The final pellets were centrifuged twice at 17,000× *g* at 4 °C for 20 min and resuspended in Tris-HCl buffer (pH 7.4). Synaptosomal protein concentrations were determined using a bicinchoninic acid assay (Pierce, Rockford, IL, USA).

### 2.8. Western Blotting

The method used to explore monomeric SNARE abundance in the hippocampus was previously described [20]. SNARE complex formation was tested in unboiled protein samples to retain high-molecular-weight complexes. The binding of primary antibodies was detected using a fluorescently-labeled secondary antibody (1:10,000) and was visualized by scanning the membranes in an Odyssey infrared imaging system (both from LI-COR Biosciences, Lincoln, NE, USA). For densitometric analysis, the signal intensity was quantified as a ratio of target protein/actin and normalized to the values of the corresponding control animals, as described in our previous reports [7,21,22,23]. The method used to explore monomeric SNARE levels in the hippocampus was previously described [20,24,25]. SNARE complex formation was tested in unboiled protein samples to retain high-molecular-weight complexes. SNARE complex formation was defined by the presence of SNAP25 immunoreactive material > 50 kDa that was absent in boiled samples [24,25]. To assess the protein content in synaptosomes, 40 µg of protein per lane was loaded on 10% gel for sodium dodecyl sulfate-polyacrylamide gel electrophoresis. After transfer onto polyvinylidene fluoride membranes, the following primary antibodies were used: anti-syntaxin 1A [1:1000; 13002S; Cell Signaling Technology (CST); Danvers, MA, USA], anti-SNAP25 (1:1000; 5308S, CST), anti-VAMP2 (1:1000; 13508S, CST), anti-synaptophysin (1:1000; 12270S, CST), anti-munc18-1 (1:1000; 13414S, CST), anti-complexin-1/2 (1:1000; AF7787; R&D Systems, Minneapolis, MN, USA), anti-CSP-α (1:1000; GTX6603; Genetex, San Antonio, Texas, USA), anti-NSF (1:1000; 3924, CST), anti-p62 (1:1000; 5114, CST), anti-Beclin-1 (1:1000; 3495, CST), and anti-GAPDH (1:1000; C1313-100; Applygen, Beijing, China). Membranes were then incubated with fluorescence-labeled secondary antibodies (1:10,000; LI-COR Biosciences, Lincoln, NE) and immunoreactivity was visualized by scanning the membranes with an Odyssey Infrared Imaging System (LI-COR). All experiments were performed by skilled technicians who were blind to sample treatment. Original images are seen in Appendix A.

### 2.9. Evaluation of Neurotransmitter Levels

The neurotransmitters norepinephrine, dopamine, and 5-hydroxytryptamine were detected in hippocampal synaptosomes by high-performance liquid chromatography with electrochemical detection, as previously described [7]. Briefly, the six channel detector potentials were set at +50, 100, 200, 300, 400, and 500 mV using a glass carbon electrode and Ag/AgCl reference electrode. The mobile phase was delivered at a flow rate of 1 mL/min at 22 °C onto the reverse-phase column. Ten-microliter aliquots were injected by an auto-injector and the cooling module was set at 4 °C. Data were calculated using an external standard calibration.

### 2.10. Statistical Analysis

Two-way ANOVA with a multiple comparison post hoc test was employed to analyze the results for the Morris water maze test, Western blots, neurotransmitters, and blood gas. The post hoc tests used were Tukey’s test and Bonferroni correction. The statistical analysis of experimental data and images was performed using SPSS 26.0 for Windows (SPSS, Illinois, USA) and GraphPad Prism 8.0 (GraphPad, San Diego, CA, USA). Data from neurobehavioral tests, biochemistry, neurotransmitter evaluation, and blood gas analysis are expressed as means ± SEM. A value of *p* < 0.05 was considered statistically significant.

## 3. Results 

### 3.1. Rapamycin Mitigated Neurobehavioral Deficits Induced by Surgery in Older Rats

To assess how surgery affected the spatial learning and memory of aged rats, Morris water maze tests were performed. The results revealed prolonged escape latencies in the surgery group on days 3 and 4 compared with those in the sham group (*p* < 0.05 or *p* < 0.01; Figure 1A). Rapamycin administration significantly decreased surgery-induced escape latency prolongation on days 3 and 4 (*p* < 0.05 or *p* < 0.01; Figure 1A). Swimming speeds did not differ among the four groups (Figure 1B). However, rapamycin alone did not affect escape latency (*p* > 0.05; Figure 1A). Rats subjected to surgery also exhibited shorter exploration times and fewer platform quadrant crossings than sham rats (all *p* < 0.01; Figure 1C,D). Rats subjected to surgery also exhibited fewer platform quadrant crossings than sham rats (*p* < 0.01; Figure 1D). These results indicated that surgery caused spatial learning and memory impairments and that pretreatment with rapamycin rescued these neurocognitive deficits. 

### 3.2. Rapamycin Ameliorated Dysregulation of SNARE and SNARE-Core Proteins Induced by Surgery

The surgery group had a significant decrease in high-molecular weight SNAP25 complex in their hippocampal synaptosomes compared with the sham group (*p* < 0.01; Figure 2A,C). We evaluated SNARE-core proteins and found that syntaxin 1A, SNAP25, and VAMP2 protein levels were all sharply reduced in the hippocampal synaptosomes of surgery group animals compared with those of sham group animals (all *p* < 0.01; Figure 2B,D–F). Intriguingly, rapamycin given in advance mitigated the decreased levels of both SNARE and SNARE-core proteins (all *p* < 0.01; Figure 2A–F).

### 3.3. Exploratory Laparotomy Had no Effect on the Levels of Synaptic Vesicle-Associated Proteins

Levels of the synaptic vesicle-associated protein synaptophysin in hippocampal synaptosomes were not affected after surgery (all *p* > 0.05; Figure 3A,B). Advance administration of rapamycin also had no impact on synaptophysin levels in the hippocampal synaptosomes of aged rats (*p* > 0.05; Figure 3A,B). These results indicated that surgical stress impaired presynaptic function by disrupting SNARE proteins but not synaptic vesicle-associated proteins.

### 3.4. Rapamycin Rescued Disruption of SNARE Chaperones Caused by Surgery

We analyzed SNARE chaperones in the hippocampal synaptosomes of aged rats. Levels of SNARE chaperones, Munc-18, complexins 1 and 2, CSPα, and NSF were all significantly decreased after surgical stress (all *p* < 0.01; Figure 4A–F). Although rapamycin could reverse the observed decreases in levels of these co-chaperones (all *p* < 0.01; Figure 4A–F), there were no significant differences in their levels among the sham, sham+rapamycin, or surgery+rapamycin groups (all *p* > 0.05; Figure 4A–F). 

### 3.5. Rapamycin Ameliorated Suppressed Autophagy in the Hippocampus of Aged Rats

We examined the effect of rapamycin on hippocampal autophagy. The levels of p62 were increased and those of Beclin-1 were decreased significantly (all *p* < 0.01; Figure 5A–C) after surgery. Rapamycin pretreatment significantly reversed the suppressed hippocampal autophagy in surgery+rapamycin group rats (all *p* < 0.01; Figure 5A–C), which was in accordance with our previous studies [7,18]. 

### 3.6. Rapamycin Reverses Surgical Stress-Induced Imbalances of Neurotransmitters

Neurotransmitter imbalance leads to cognitive dysfunction and is involved in dNCR [7]. Neurotransmitter release is closely related to SNARE proteins and their chaperones. Evaluation of a series of neurotransmitters by high-performance liquid chromatography showed that norepinephrine was upregulated in the synaptosomes of surgery group rats compared with that in the synaptosomes of sham group rats (*p* < 0.01; Figure 6A). In contrast, both dopamine and 5-hydroxytryptamine levels were downregulated in the synaptosomes of surgery group rats (all *p* < 0.01; Figure 6B,C). Rapamycin significantly reversed the imbalances of these three neurotransmitters in surgery+rapamycin rats (all *p* < 0.01; Figure 6A–C), which was consistent with our previous studies [7]. 

### 3.7. Arterial Gas and Glucose Analysis

No significant differences in blood oxygen or glucose concentrations were observed among the groups in this study (Figure 7). Therefore, our findings verified that neurobehavioral impairments, synaptic vesicle function deficit, and stress-induced imbalances of neurotransmitters are not likely to be caused by hypoxia, hypercapnia, or hypoglycemia in the hippocampus.

## 4. Discussion

The current findings indicate that exploratory laparotomy decreased levels of SNARE monomers and structural proteins and disturbed neurotransmitter release in hippocampal synaptosomes, ultimately resulting in learning and memory deficits in aged rats. The protective effect of rapamycin against learning and memory impairment induced by surgery occurred in part through the amelioration of structural and functional SNARE abnormalities. 

Current animal models of postoperative cognitive impairment are mostly based on inhalation anesthetics, such as isoflurane or sevoflurane [19,26]. Propofol is currently the most commonly used intravenous anesthetic in clinical practice. However, there can still be deficits in postoperative cognitive functions following propofol administration. Lian et al. found that postoperative cognitive dysfunction was alleviated in 20-month-old male Fischer rats that underwent cardiac surgery with anesthesia, using 15 mg/kg propofol for induction and 1 mg/kg/min propofol for maintenance [27]. Mardini et al. found that 9–14-month-old 3xTgAD Alzheimer transgenic mice given one injection of propofol (250 mg/kg) with or without cecal ligation and excision displayed minimal to no changes in short- and long-term behavior and no changes in neuropathology [28]. Lee et al. demonstrated that 18-month-old rats that received propofol infusion alone (0.6 ± 0.1 mg/kg/min) for 2 h also exhibited no spatial memory impairment [29]. However, postoperative cognitive deficits have been observed following propofol anesthesia in several studies. Zhang et al. reported that 20-month-old male Fischer rats received intravenous 15 mg/kg propofol to induce anesthesia followed by 1 mg/kg/min propofol to maintain anesthesia and then underwent right carotid exposure surgery. The surgical procedure lasted for 15 min and the propofol anesthesia lasted for 2 h. The propofol-based anesthesia with carotid exposure induced a similar degree of neuroinflammation and learning and memory impairment compared with 2 h isoflurane anesthesia with the same surgical stress [30]. Liu et al. reported that aged male Sprague–Dawley rats (18–20 months old) that received intraperitoneal propofol injections of 200 mg/kg once a day or every 9 days, 6 times, had neuronal damage and cognitive impairment [30]. Memory impairment has also been reported in 4–6 week-old Sprague–Dawley rats that received one injection of 10 mg/kg propofol [31]. We previously showed that 4 h propofol anesthesia alone and 2 h propofol anesthesia plus laparotomy surgery can induce early cognitive impairments, while propofol anesthesia for 2 h alone had no effect on cognition [7,18]. The reasons underlying these discrepancies are very complex. It can be concluded that preoperative states, such as low cognitive reserve, surgical stress, and prolonged exposure to anesthesia, can promote postoperative cognitive impairment. In the present study, learning and memory impairments were induced when combined with laparotomy. However, the effect of short-term propofol anesthesia on neurocognitive function is not obvious. Furthermore, no alternative injectable anesthetic, such as etomidate, was used for the sham surgery. Therefore, we can only suggest that the surgical trauma is the principle factor of dNCR induction.

In synaptosomes, correct neurotransmitter release depends on the normal function of Ca^2+^ sensors and correct assembly of the SNARE complex. Accumulating evidence demonstrates that abnormal abundance or dysfunction of the SNARE complex in synaptosomes contributes to disordered neurotransmission and, ultimately, synaptic dysfunction correlated with neurodegenerative, neuropsychiatric, and neurodevelopmental diseases [32,33]. Inhibition of SNARE complex formation results in defects of SNARE-dependent exocytosis and abnormal regulation of SNARE-mediated vesicle fusion, which are related to the pathogenesis of neurodegenerative diseases [34]. Levels of syntaxin 1, SNAP25, and VAMP2 were all reduced in cerebral samples of amyloid precursor protein transgenic mice [35]. Because VAMP2 is involved in both synaptic vesicle exocytosis and endocytosis [36,37], its loss also leads to neurodegeneration [38]. Furthermore, perturbations of its N-terminal domain can inhibit neurotransmitter release [39]. Syntaxin 1A reportedly interacted specifically with intracellular β-amyloid monomers and oligomers in Alzheimer’s disease pathogenesis [40]. Delayed neurocognitive recovery can also involve pathological changes of cerebral β-amyloid accumulation, the most typical pathophysiology of Alzheimer’s disease [41]. However, no previous investigation has evaluated the role of SNARE proteins in dNCR. In the present study, we found that surgery significantly downregulated the levels of three core proteins (SNAP25, VAMP2, and syntaxin 1) in the SNARE complex and inhibited SNARE complex formation, indicating that disruption of SNARE-induced neurotransmission underlies the pathology of dNCR. 

Co-chaperones, such as Munc18, complexins 1 and 2, and α-synuclein make SNARE assembly and disassembly more efficient [9,10,17]. The Sec1/Munc-18-like protein, Munc18-1, is a molecular chaperone of syntaxin 1, which is involved in SNARE-mediated membrane docking and fusion [42]. Complexins 1 and 2 act as chaperones by binding to SNAP25. NSF is vital for membrane fusion and requires the formation of SNARE proteins from target and vesicle membranes. After fusion, the SNARE complex is dissociated by the ATPase NSF for further rounds of fusion [9]. CSPα is an essential molecular co-chaperone that promotes SNARE complex assembly by chaperoning SNAP25 during synaptic activity [43]. In the present study, the levels of all these SNARE chaperones were significantly reduced, consistent with the observed decrease in SNARE formation protein levels, indicating disrupted SNARE structure and functionality in the hippocampal synaptosomes. In contrast, we previously observed profound α-synuclein oligomer aggregation, probably because inhibition of hippocampal autophagy sharply weakened α-synuclein oligomer clearance. Rapamycin ameliorated both structural and functional impairments of the SNARE complex, indicating that under anesthesia/surgical conditions, SNARE complex structure and function are closely related to autophagy. Interestingly, the synaptic marker synaptophysin was unaffected, revealing that disturbances of SNARE proteins and their chaperones in hippocampal synaptosomes—not synaptic vesicle loss—led to dNCR. 

The mammalian target of rapamycin (mTOR) signaling pathway is a master regulator of cell growth and metabolism [44]. It has two functionally distinct complexes, mTOR complex 1 (mTORC1) and mTOR complex 2 (mTORC2). mTORC1 is a signal integrator that responds to multiple signals, such as growth factors, nutrients, energy, and oxygen status, to control cell growth and proliferation processes, including mRNA biogenesis, protein, lipid and nucleotide synthesis, energy metabolism, and autophagy [45]. mTORC2 participates in cell survival and actin cytoskeleton organization [45]. Rapamycin, a specific inhibitor of mTORC1, has broad effects on physiological processes [46]; however, rapamycin does not completely inhibit mTORC1-mediated processes, such as protein synthesis and autophagy. More importantly, rapamycin has immunosuppressive effects. This may explain why in Figure 1D there is a negative effect of rapamycin in the sham+rapamycin group compared with the findings in the sham group, which is different from our previous studies [18]. This is why rapamycin has some limitations, despite its use in treating diseases such as cancer, diabetes, obesity, neurological diseases, and genetic disorders. For example, the clinical success of rapamycin has been associated with only a few benign and malignant cancers [46]. Moreover, clinical trials have demonstrated that while rapamycin treatment can induce tumor regression, tumors regrow upon cessation of treatment [47]. More importantly, mTOR inhibitors (sirolimus, everolimus) are also widely used as immunosuppressive agents to prevent organ rejection after organ transplantation, but they can also cause many side effects, such as hypertension, infection, and hyperlipidemia [48].

Aberrant SNARE complex structure and function will inevitably cause abnormal release of neurotransmitters, which is “the final stage” of dNCR. We found that surgical stress altered the levels of three neurotransmitters (norepinephrine, dopamine, and 5-hydroxytryptamine) in hippocampal synaptosomes, while rapamycin restored neurotransmitter equilibrium and further improved cognitive behavior, which was consistent with our previous findings [7]. However, how the SNARE complex regulates the release of specific neurotransmitters after surgery remains undetermined.

The SNARE complex is absolutely necessary to mediate vesicular fusion events during autophagy, including autophagosome formation and maturation, autophagosome–lysosome fusion, autolysosome formation, and other processes [12,49]. Therefore, autophagy and SNARE complex functions are closely interrelated. We speculate that surgical stress induces the inhibition of autophagy followed by the abnormal accumulation of toxic aggregates, such as α-synuclein oligomers (as indicated in our previous report [7]), which impair SNARE structure and function. The inhibition of autophagy and the disruption of SNARE structure and function in synaptosomes are mutually causal and can result in feed-forward regulation that triggers imbalanced neurotransmitter release, ultimately resulting in postoperative cognitive impairment. Rapamycin partially restored the structure and function of the SNARE complex by scavenging accumulated toxic substances as a result of enhanced autophagy (Figure 8).

There were several limitations to the current study. First, the experiment did not represent a clinical setting and the inhalation anesthetics sevoflurane and isoflurane have sedative as well as analgesic and muscle-relaxant effects [50]. However, the intravenous anesthetic propofol only has a sedative effect [51]. Therefore, it was not appropriate that we used propofol alone to perform an extremely invasive surgery in rats without an analgesic before and during surgery. Although oxybuprocaine hydrochloride gel was applied topically after surgery every 8 h for 48 h, this may have been inadequate. The loss of consciousness does not prevent the transmission of pain stimuli and processing by the central nervous system [52]. Moreover, the SNARE complex is also involved in the nociceptive pathway [53,54,55] and this will have definitely affected our results. Second, this study cannot fully explain the effect of propofol on postoperative cognitive function because no alternative injectable anesthetics were used for the sham group. Moreover, SNARE proteins are targets of general anesthetics, including propofol [56,57,58,59]. 

Third, we used laparotomy to generate the dNCR model. Accumulating evidence supports the gut–brain axis as being extremely important for maintaining the function of the central nervous system (CNS) and intestinal dysbiosis can affect cognitive function during multiple chronic CNS diseases [60]. Although we did not explore the intestinal environment, our previous study indicated that the intestinal microbiota can affect cognitive function by affecting the activation of central microglia in APP/PS1 mice [61]. Therefore, a new research direction may be to explore the mechanism by which the intestinal environment affects the synaptic vesicle function of neurons. Fourth, open-field fear conditioning and radial arm maze tests may be more suitable for animals 24 h post-surgery compared with the Morris water maze test, which requires animals to be put into water. Lastly, because of animal number limitations and the experimental design, each group only had four samples for Western blot analysis.

## 5. Conclusions

Our study demonstrated experimentally that surgery induced abnormal SNARE function and disrupted neurotransmitter homeostasis in hippocampal synaptosomes, possibly by local suppression of autophagy. These interrelated and causal pathological events may ultimately promote the occurrence and development of dNCR (Figure 8). Our findings contribute to elucidating the molecular synaptic mechanisms underpinning postoperative neurological disorders. Moreover, drugs capable of improving SNARE complex structure and function, such as rapamycin, could be harnessed to combat postoperative cognitive impairments (Figure 8).

## Figures and Tables

**Figure 1 brainsci-13-00598-f001:**
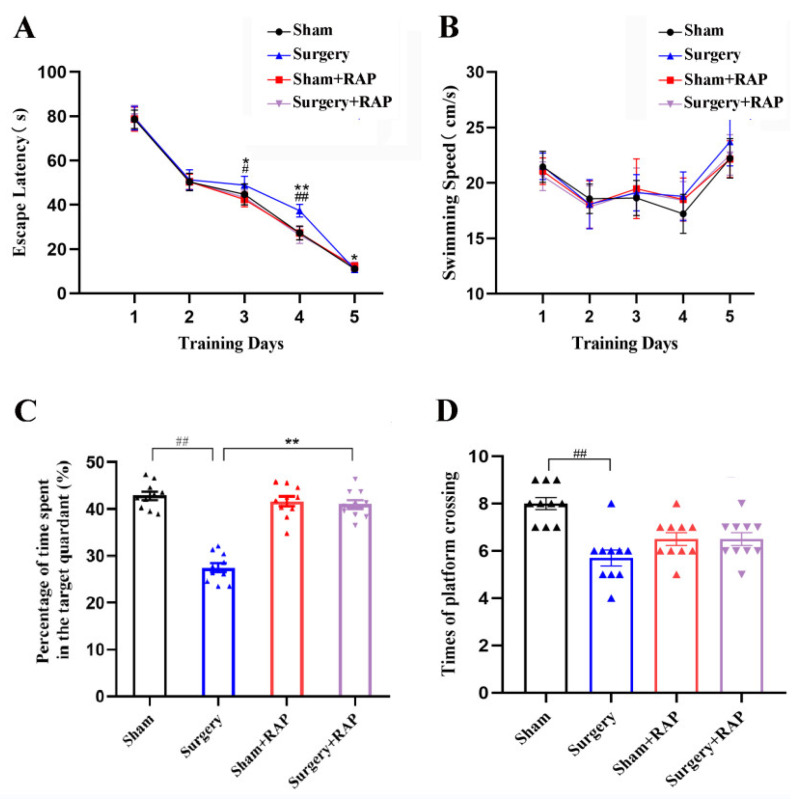
Rapamycin mitigated surgery-induced spatial learning and memory impairments in elderly rats. (**A**) Surgery increased escape latency on days 3 and 4 compared with sham treatment. Rapamycin administration ameliorated surgery-induced escape latency prolongation, but rapamycin alone did not affect escape latency. (**B**) Neither surgery nor rapamycin affected swimming speed. (**C**,**D**) Surgery decreased both the percentage of dwell time in the target quadrant and number of platform crossings compared with sham treatment. Rapamycin attenuated surgery-induced decreases in the percentage of dwell time in the target quadrant. Values represent the mean ± SEM, n = 10 per group. ^#^
*p* < 0.05, ^##^
*p* < 0.01 surgery vs. sham group. * *p* < 0.05, ** *p* < 0.01 surgery+rapamycin vs. surgery group. RAP: rapamycin.

**Figure 2 brainsci-13-00598-f002:**
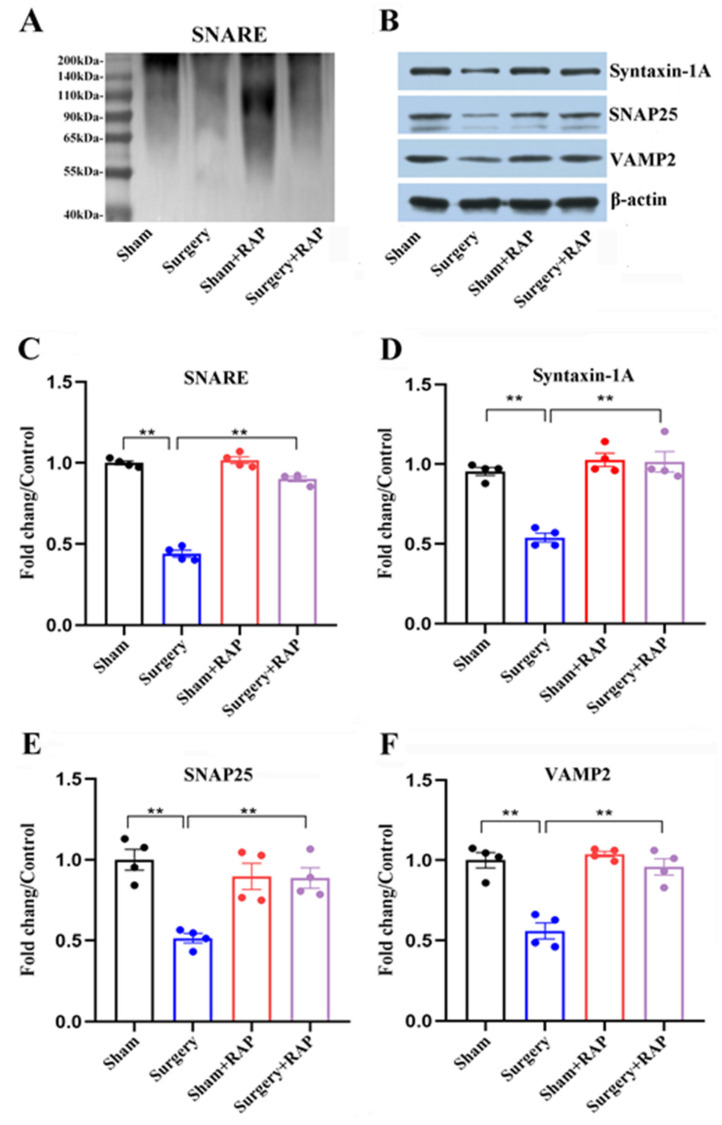
Rapamycin attenuated disturbances of SNARE proteins in synaptosomes following surgery. (**A**,**B**) Representative immunoblots illustrating SNARE (**A**) and its core proteins syntaxin 1A, SNAP25, and VAMP2 (**B**) in hippocampal synaptosomes. (**C**) Decreased SNARE complex levels in the surgery group compared with those in the sham group were reversed by rapamycin. (**D**–**F**) Rapamycin ameliorated surgery-induced decreases in levels of three core SNARE complex proteins: syntaxin 1A (**D**), SNAP25 (**E**), and VAMP2 (**F**). Values represent the mean ± SEM, n = 4 per group. ** *p* < 0.01 surgery vs. sham group or surgery+rapamycin vs. surgery group. RAP: rapamycin.

**Figure 3 brainsci-13-00598-f003:**
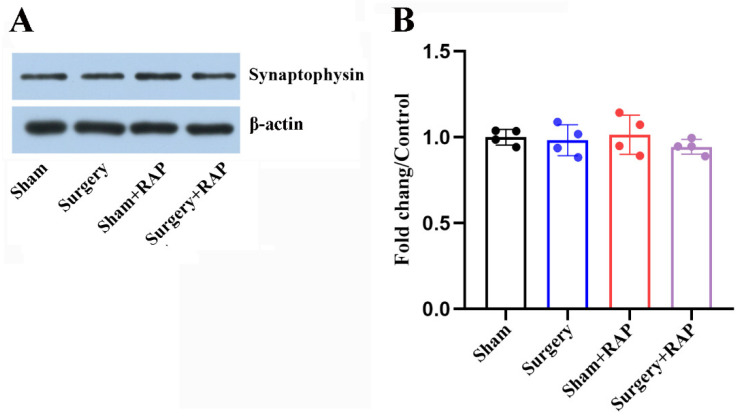
Surgery did not impact synaptic vesicle markers in hippocampal synaptosomes. (**A**) Representative immunoblots of synaptophysin in hippocampal synaptosomes. (**B**) No differences in synaptophysin levels were detected among the four groups (all *p* > 0.05). Values represent the mean ± SEM, n = 4 per group. RAP: rapamycin.

**Figure 4 brainsci-13-00598-f004:**
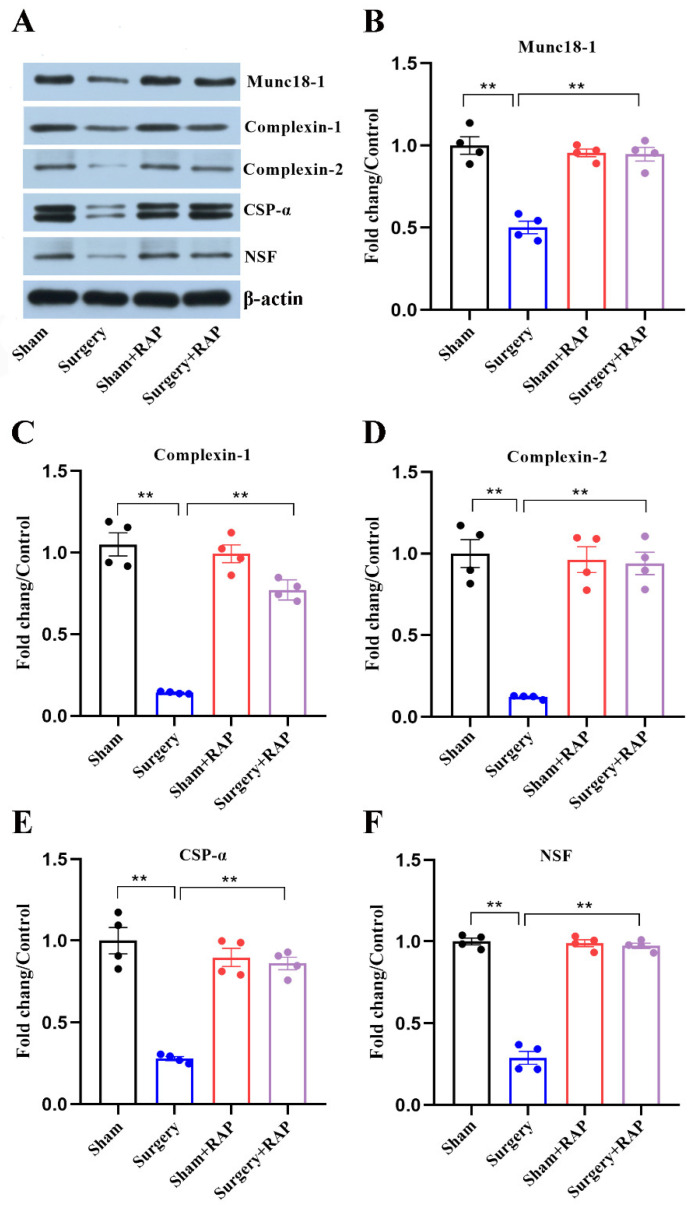
Rapamycin alleviated disturbances in the levels of SNARE chaperones in hippocampal synaptosomes caused by surgery. (**A**) Representative immunoblot of SNARE chaperones in hippocampal synaptosomes. (**B**) Munc18-1, (**C**) complexin 1, (**D**) complexin 2, (**E**) CSPα, and (**F**) NSF were quantified by densitometry and normalized to GAPDH. Values represent the mean ± SEM, n = 4 per group. ** *p* < 0.01 surgery vs. sham group or surgery+rapamycin vs. surgery group. RAP: rapamycin.

**Figure 5 brainsci-13-00598-f005:**
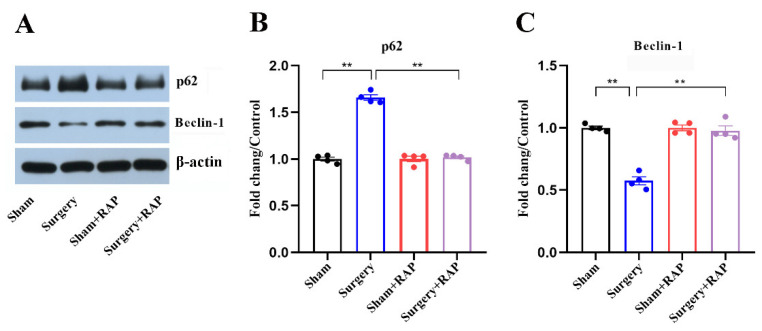
Rapamycin rescued the inhibited autophagy in hippocampal synaptosomes of aged rats. (**A**) Representative immunoblots illustrating the effects of rapamycin on surgery-induced changes in hippocampal autophagy-related protein levels. (**B**) Rapamycin reversed the surgical stress-induced increase in p62 levels and decrease in Beclin-1 levels (**C**) in hippocampal synaptosomes. Values represent the mean ± SEM, n = 4 per group. ** *p* < 0.01 surgery vs. sham group or surgery+rapamycin vs. surgery group. RAP: rapamycin.

**Figure 6 brainsci-13-00598-f006:**
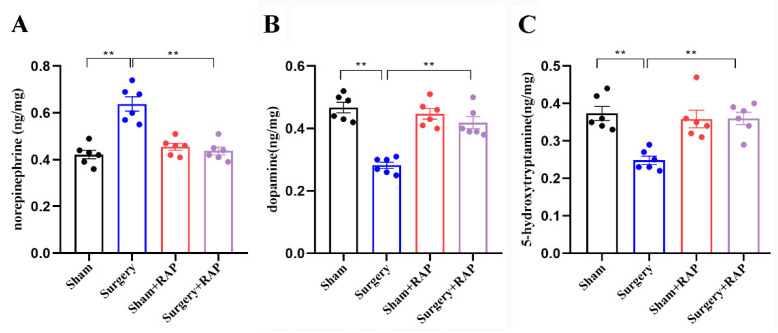
Rapamycin reversed surgical stress-induced imbalances of neurotransmitters in hippocampal synaptosomes. Levels of norepinephrine (**A**) were upregulated in synaptosomes of rats that underwent surgery compared with those in synaptosomes of sham rats. Dopamine (**B**) and 5-hydroxytryptamine (**C**) were downregulated in synaptosomes of rats that underwent surgery. n = 6 rats per group. ** *p* < 0.01 surgery vs. sham group or surgery+rapamycin vs. surgery group. RAP: rapamycin.

**Figure 7 brainsci-13-00598-f007:**
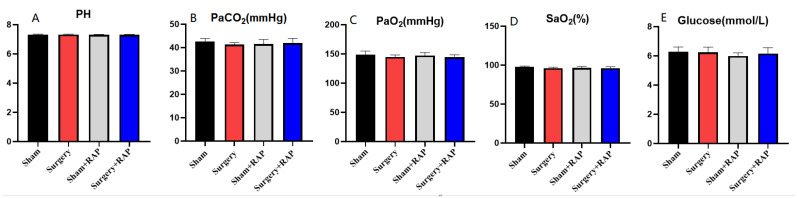
Exploratory laparotomy had no significant effect on blood gas or blood glucose in aged rats and was not influenced by rapamycin pretreatment. (**A**) There was no difference in the PH value of the four groups. (**B**) There was no difference in the partial pressure of carbon dioxide between the four groups. (**C**) There was no difference in oxygen partial pressure between the four groups. (**D**) There was no difference in blood oxygen saturation between the four groups. (**E**) There was no difference in blood glucose between four groups. PaCO_2_, partial pressure of carbon dioxide in arterial blood; PaO_2_, partial pressure of oxygen in arterial blood; SaO_2_, arterial oxygen saturation; Hb, hemoglobin; Glucose, blood glucose. Values represent the mean ± SEM (n = 5/group). RAP: rapamycin.

**Figure 8 brainsci-13-00598-f008:**
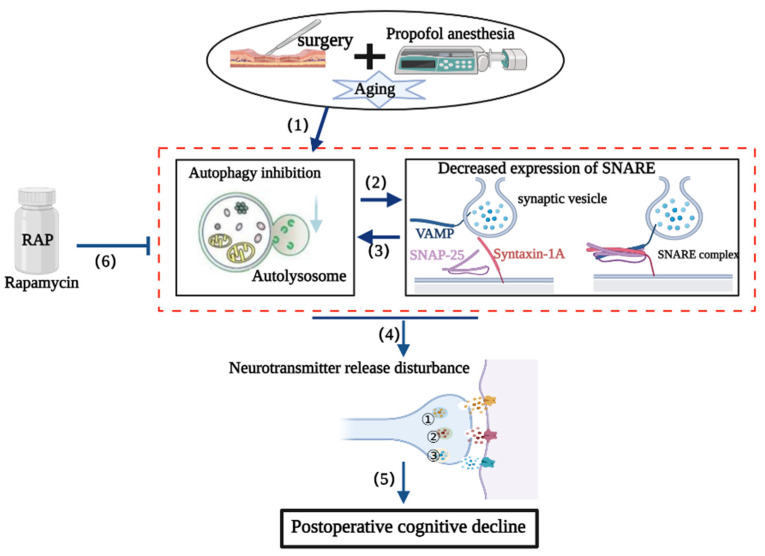
Possible mechanism of SNARE complex disturbances in the delayed neurocognitive recovery (dNCR) pathological process. (1) Surgery can inhibit autophagy in the hippocampus. (2) Hippocampal autophagy inhibition further exacerbates the structure and function of the SNARE complex. (3) SNARE complex dysfunction further aggravates the impairment of autophagy. (4) SNARE complex disturbances and disrupted hippocampal autophagy impair neurotransmitter release, ultimately promoting delayed neurocognitive dysfunction (5). (6) Rapamycin can ameliorate postoperative neurocognitive deficits, probably by restoring suppressed autophagy and then improving the structure and function of the SNARE complex. ① norepinephrine; ② dopamine; ③ 5-hydroxytryptamine.

## Data Availability

The raw data supporting the conclusions of this article will be made available by the authors, on reasonable request.

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
