# Peer review of "Rapamycin Affects the Hippocampal SNARE Complex to Alleviate Cognitive Dysfunction Induced by Surgery in Aged Rats"

_brainsci, 2023, doi:10.3390/brainsci13040598_

Round 1

Reviewer 1 Report

In this study, Kang and colleagues aimed to evaluate the effect of “propofol anesthesia” with surgery in SNARE proteins and chaperones in hippocampal synaptosomes and its impact in learning and memory behavior, and the effect of rapamycin in postoperative recovery using rats as an animal model.
Hence, Kang and colleagues performed an exploratory laparotomy, using an IP injection 50mg/kg of propofol following an IV of 0.9mg/kg/min of propofol. After the surgery, oxybuprocaine hydrochloride gel was applied topically every 8h for 48h. To study the rapamycin effect in these animals, two groups of animals, sham+rapamycin and surgery+rapamycin, were treat with 10mg/kg/day for 7 consecutive days until the surgery day.
Water maze test, a behavior test to access special memory and learning, was conducted 24h after the surgery.
For sample collection, animals were exposed to 6% of isoflurane for 60s and decapitated 3 days after surgery.
Furthermore, the authors stated, “Utmost efforts were made to minimize the number and suffering of the experimental animals”.
The results suggested that the surgery group showed impairment of learning and memory, decreased expression of SNARE, SNARE core proteins, SNARE chaperones in hippocampal synaptosomes, inhibition of autophagy and neurotransmitters imbalance. However, rats treated with rapamycin before the surgery seem protected from these abnormalities.

Comment:
This study is potentially,  interesting, however, I regret to inform you that the whole investigation and results are compromised due to the suffering of the experimental animals and its impact in the research outcome.
1.    Why using propofol for a major surgery?! In major surgeries, e.g. laparotomy, anesthetics and analgesics are essential since the procedure results in physical and physiological injury. The authors used propofol alone to perform an extremely invasive surgery in rats without providing them with an analgesic, before, during and after the surgery. The loss of conscience does not prevent the transmission of the pain stimuli and its processing by the central nervous system.
2.    Human anesthesia does not involve solely propofol for any kind of invasive surgery. Once again, the usage of only propofol is not justified. This experiment does not represent clinical settings.
3.    The post-operative wellbeing of the animals was neglected. Analgesic procedures should be initiated when the symptoms of pain are perceived in the animal. Behavioral and physiological changes occur when the animal feels pain.
4.    After surgery, animals should be keep in a clean location with an appropriate temperature and humidity. However, the experimental animals were subjected to MWM 24h after the surgery. Was this test adequate considering the physical capacity of the animal 24h after an invasive surgery? Especially in absence of analgesics?
5.    SNARE complex is also involved in nociceptive pathway. Considering that the animals were feeling post-surgical pain, how can the authors tell the causative agent of these molecular and behavioral changes?

Reviewer 2 Report

Kang et al. Rapamycin Affects the Hippocampal SNARE Complex to Alle-2 viate Cognitive Dysfunction Induced by Propofol Anesthesia 3 Plus Surgery in Aged Rats

The authors use a rat model to evaluate the effects of propofol anaesthesia + surgery on learning and memory and make a link between poor performance on a Morris water maze test and altered expression of key SNARE proteins. They postulate that these changes in SNARE protein expression or function is the cause of delayed neurocognitive recovery in elderly individuals.

The authors found that rapamycin, an autophagy enhancer, could ameliorate the effects of surgery and propofol on the rat’s performance in a Morris water maze test. Furthermore, the expression of SNARE proteins (syntaxin 1A, SNAP25 and VAMP2) and SNARE chaperone proteins (Munc-18, Complexin-1, and Complexin-2, CSP-α, and NSF) were all reduced in hippocampal synaptosomes of surgical animals compared with sham group animals. They found that rapamycin could return expression of these proteins to sham levels. In contrast, synaptophysin, a synaptic vesicle-associated protein had no expression change in any of the groups indicating that synaptic vesicles were not lost during propofol treatment. Intravenous propofol anaesthesia combined with exploratory laparotomy did not influence blood gas or blood glucose concentrations.

SNARE complex proteins have previously been shown to be targets of general anaesthetics, including propofol. Therefore, it could make sense that expression or function of these presynaptic proteins are affected by the procedure. However, a critical reader might be puzzled by the link to autophagy and endocytosis and a potential mechanism. My understanding is that autophagy and protein degradation pathways employ a completely different set of SNAREs than what is used for neurotransmission, e.g. syntaxin17 rather than syntaxin1A (see for example Wang et al, 2016, Semin Cell Dev. Biol. 60: 97-104).  If autophagy and debris clearance is implicated as the relevant mechanism here, shouldn’t the authors be examining expression of the autophagy-related SNAREs rather than the neurotransmission-related ones? Or, are they proposing a mechanisms centred on neurotransmission, in which case the autophagy angle of rapamycin seems unclear?

Additionally, the way that rapamycin overturns the effects of propofol on the expression of a wide range of proteins seems to indicate that rapamycin’s protective mechanism is occurring upstream from SNARE complex formation and does not seem to be specific to the proteins studied here.

Does rapamycin simply overturn the effects of propofol on a wide variety of protein expression? If so, it would be hard to definitively state that the changes in SNARE protein expression cause delays in neurocognitive recovery. Similarly, it would be hard to definitively state that the effect of rapamycin on SNARE protein expression causes the rescue of behaviour after propofol and surgery. Some additional controls seem to be needed here (e.g., autophagy SNAREs?), to show that this is a specific rather than a general effects, and ideally to better link the findings to the actual mechanism of action underlying rapamycin. For example, it would be interesting to include a more thorough dissection of rapamycin’s mTOR pathway and how this could be linked to the downregulation of the proteins after propofol treatment. Nevertheless, the correlation is interesting and opens new possibilities for using rapamycin to ameliorate the effects of general anaesthetics on the aged brain.

It is also necessary to clarify how all the immunoblots were quantified and ideally to show all of the immunoblots involved in this study, for example in supplementary data.  The n’s are fairly small (e.g., n=4) so this should be easy to organise as a set of supplemental figures.

The role of inflammation, and rapamycin as a potential anti-inflammatory drug, has not been adequately discussed. In other words, this may have more to do with the surgical intervention than with anaesthesia, and the adequacy of the sham control group in Fig. 1 isn’t quite clear.

Minor:

Line 41: Why is there a difference between cardiac and non-cardiac surgery?

Line 102: Should it be syringe pump?

Line 138: Grammar. Suggested edit: “Carbon dust, an edible pigment, was used to darken the swimming water”

Line 178: glass, not glassy

Line 119: In the sham group, were the rats shaved and incised? 

Figure 1C: Error bars are unclear.

Figure 2A: It is unclear which part of the band was quantified. Please include an image of the full blot. Please include your quantification methods for this and other blots (see above).

Line 253-256: The phrasing here is a bit confusing.

Line 265: Could you briefly explain how p62 and Beclin-1 expression are indicators of hippocampal autophagy.

Table 1: Would perhaps be clearer as a graph.

Line 370: Unclear what is meant by “last kilometre”

Line 406: Consider combining the sentences discussing laparotomy and the gut-brain axis.

Reviewer 3 Report

In this article, the authors attempt to show that surgery induced cognitive dysfunction is improved by rapamycin through its actions on the hippocampal SNARE complex. Overall, the authors present a fairly straightforward study linking surgical manipulation of the small intestine with changes in protein expression and neurotransmitters in the hippocampus, suggesting a negative impact on synaptic plasticity with downstream effects on cognition. Nevertheless, the lack of stringency for the statistical analyses and the insistence by the authors to link the effect to propofol when that variable was not tested dampens my enthusiasm for this study.

Concerns:

1)      Both the title and the statement “we evaluated the effects of propofol anesthesia plus surgery on learning and memory” in the abstract suggest that the authors are assessing the interaction between surgery and the use of propofol, which is not the case. The sham surgery has the same conditions, including anesthesia. If the authors were assessing the effect of propofol, then, at a minimum, an alternative injectable anesthetic should have been used for the sham condition. For clarity, the authors should remove any suggestion that propofol is the variable under investigation.

2)      This is a 2 x 2 design: sham +/- rapamycin and surgery +/- rapamycin. Thus, there may be a main effect of surgery, a main effect of rapamycin, or an interaction between surgery and rapamycin. However, only a one way ANOVA was used to assess groups, except in the case where time was a component and a 2 way RM ANVOA was used. A 2-WAY ANOVA with a multiple comparison post hoc test should be employed to analyze the results for the MWM, western blots, neurotransmitters, and blood gas.  

3)      Figure 1 D. Without the proper analysis it is difficult to understand the results stated in the text and shown in the graph. There is a significant decrease in the number of platform crossings in the surgery group, with a slight, though not significant, increase in animals that were treated with rapamycin, either the sham or surgery group. However, the text states “Rapamycin application also obviously increased the time spent in the platform area compared with surgery alone 202 (p < 0.01; Figure 1D).”

4)      While the experimental groups and division of rats were explained in the methods, how they were divided for analyses is not clear and does not add up. The starting # was 20/group. 5/group were used for blood analysis and immediately euthanized following surgery. That left 15/group. MWM shows only 10/group, while the westerns blots are 4/group, and then 6/group for the neurotransmitter analysis. Where are the remining 5 rats? Why were so few animals used for western blot analysis? Couldn’t 1 hippocampus have been used for westerns while the other was used for neurotransmitters (increasing the n)?

5)      Although the results seem very straight forward for the western data, 4/group is still low and likely does not have enough power to discern significant differences for small effects (e.g. SNARE and complexin-1 for surgery + RAP does not fully recover to sham levels).

6)      Discussion. The authors elaborate on the potential effects of propofol on cognition as the reason for doing this study. However, as stated previously, they did not test this and their results for the sham group, which received propofol suggest that there is no negative effect. Yet, this is not discussed. Overall, there seems to be a disconnect between what the authors thought the goal of this study was and what their design actually tested.  

Reviewer 4 Report

In this paper the authors are investigating the effects of propofol anesthesia and surgery on the neurocognitive recovery in 22 months-old rats. They first demonstrate using the Morris water maze test that surgery together with propofol anesthesia led to memory impairment in old rats, which can be rescued by rapamycin injection 7 days before the surgery. Then they asked which mechanisms might be responsible for the memory loss. First, they found that within the synaptosome, SNARE proteins were downregulated following surgery and anesthesia, and that rapamycin rescued this phenotype. In contrast, they found that the synaptic vesicle associated protein synaptophysin was not affected by the surgery and anesthesia. Altogether these results suggest that only the anchor of the synaptic vesicles at the membrane before neurotransmitter release and not the number of vesicles is affected by the surgery plus anesthesia in this model. To further test this hypothesis Kang et al, measured the expression level of SNARE chaperones, which are necessary for the SNARE assembly. They found a decrease of the expression of these chaperones, rescued by the rapamycin treatment. They also found a decrease of the expression of some neurotransmitters, which was rescued by rapamycin treatment. Since rapamycin is known to promote autophagy, the authors measured the expression of autophagy related proteins and found reduced autophagy after surgery and anesthesia, that can be rescued by rapamycin treatment. No linked was further demonstrated between the autophagy phenotype and the neurotransmitter release in this manuscript, but the authors discussed their hypothesis in the discussion. Finally, the author excluded a contribution of hypoxia, hypercapnia, or hypoglycemia to the memory impairment induced by surgery and anesthesia.

The authors have done a good work to better understand how anesthesia and surgery lead to memory impairment in old rat. The structure of the manuscript overall is good, the introduction, results and discussion are satisfactory. However, the syntax and grammar are not meeting the scientific standards for publication, which make the paper hard to read for now. For this article to be finalized, I suggest the following revisions:

Major:

1.     The syntax and grammar through the entire manuscript do not meet scientific standards for publication. The authors need to fix it to help the readers. Here are just few examples:

-        l.16 “in synapses”, l.18 “can aggregated..”, l.22 “Aged rats administrated propofol anesthesia and surgery exhibited”, l.42 “dNCR has…, because they…”

-        l.72, “indeed” instead of “therefore”?

-        l.73 “thereby regulating the entirety of neurotransmitter release”, I’m not sure what the authors mean by entirety in this context, should you just write “regulating the neurotransmitters release”?

-        l.77, “bring about”, rather “lead to”?

-        l.146 “according to our previous research” rather “following our previously reported protocol”

-        l.204 “could bring about spatial” etc.

-        l.268 “which is accordance” should be “which is in accordance”

2.     The authors should edit their conclusions l.298-299 “does not lead to neurodegenerative 298 pathologies in the hippocampus resulting from hypoxia, hypercapnia, or hypoglycemia”. Indeed, this experiment do not allow to conclude there is no neurodegenerative pathologies observed in the brain. It just suggests that behavioral impairments, synaptic vesicle function deficit and Stress-Induced Imbalances of Neurotransmitters phenotypes are likely not due to hypoxia, hypercapnia, or hypoglycemia.

3.     In the discussion, l.337, the authors claim that “decreased SNAP25 is a typical marker for neuronal loss”, it is not the most typical marker for neuronal loss. The assessment of the neuronal loss is not covered by this article and would require other experiments, for example some histological staining for neuronal cells marker (MAP2). But this is out of the scope of this manuscript. The authors should however modify their conclusions.

4.     In the discussion, the authors should comment on Fig1D negative effect of rapamycin and comment on the limitation of treated patients with rapamycin in the paragraph l363 of their discussion for example. 

Minor:

1.     Results part 3.2 would beneficiate from a 1st sentence introduction.

2.     Fig1C. significant bar have been shifted to the right

3.     Fig2A, the author could label SNARE on top of the immunoblot

Round 2

Reviewer 1 Report

Comment in the file. 

Author Response

A sutured wound heals in 24h? There are other behavior tests that could be used to

access the spartial memory of rodents, e.g. radial arm maze, which would be more

suitable in this case, than placing sutured animals in water.

Response (2nd): Thank you for this important question. It is more suitable for animals to perform open field test, fear conditioning test, Y maze test or radial arm maze than  the MWM 24 hours after the operation. We have added this limitation in the revised manuscript (Page 14-15, line 463-465). we will further improve our experimental methodology in our subsequent studies. Thank you very much for this important comment again.

Round 3

Reviewer 3 Report

Thank you for addressing my concerns and considering improvements for future investigations.